Effects of 440-Hz vs. 432-Hz preferred music frequencies, during warm-up, on intermittent anaerobic speed test performance in men and women kickboxers: a double-blind crossover study

Jebabli Nidhal nidhaljebali@issepks.u-manouba.tn 1
Boujabli Manar 1
Khlifi Mariem 1
Ouerghi Nejmeddine 1
Bouassida Anissa 1
Ben Abderrahman Abderraouf 2
van den Tillaar Roland roland.v.tillaar@nord.no 3
1 Research Unit: Sport Sciences, Health and Movement, High Institute of Sport and Physical Education of Kef, University of Jendouba , Kef , Tunisia
2 Higher Institute of Sport and Physical Education of Ksar-Said, University of Manouba , Manouba , Tunisia
3 Department of Sports Science and Physical Education, Nord University , Levanger , Norway
Coswig Victor
Electronic publication date: 2025 Mar 6
Publication date: 2025
Volume: 13
Electronic Location ID: e19084
Received 2024 Nov 6; Accepted 2025 Feb 10
Copyright: ©2025 Jebabli et al.
Copyright year: 2025
Copyright holder: Jebabli et al.
License: This is an open access article distributed under the terms of the Creative Commons Attribution License, which permits unrestricted use, distribution, reproduction and adaptation in any medium and for any purpose provided that it is properly attributed. For attribution, the original author(s), title, publication source (PeerJ) and either DOI or URL of the article must be cited.
License URL: https://creativecommons.org/licenses/by/4.0/

Keywords: Fast tempo music, Tone frequencies, Specific exercise, Psychophysiological responses, Physical performance

Funding: No sources of funding were used to assist in the preparation of this article.

==============================
Background

Preferred music has been shown to enhance psychological and physiological parameters in order to increase physical performance in high intensity exercises. However, the effects of preferred music are less conclusive with different frequencies. The present study assessed the effects of listening to preferred music during warm-up at different frequencies on physical performance and psychophysiological responses specific in male and female kickboxers.

Methods

In a double-blind crossover study design, fifteen men and thirteen women kickboxers randomly performed the intermittent kickboxing anaerobic speed test (IKAST) after listening to preferred music around 440 Hz (PM44Hz), or 432 Hz (PM432Hz) frequencies or no music during warm-up. Physical performance indices, heart rate, blood lactate, rating of perceived exertion (RPE) and feeling scale (FS) were measured.

Results

Warm-up with PM440 Hz significantly improved IKAST performance indices with the highest impact velocity and FS, lowest mean heart rate and RPE, followed by PM432 Hz for both sexes compared to the control condition. For sex interaction, men had lower heart rate with PM440Hz, women the lowest with PM432Hz. In addition, women had higher positive feeling scale with PM440Hz, while men did not experience any significant change between the two musical conditions.

Conclusion

PM440Hz during warm-up was found to be more effective in improving specific performance, positive mood with a potential dissociation from discomfort during the test. Also, women were more affected by the music frequency difference compared to men.

Introduction

Combat sports such as kickboxing are disciplines that are generally organized by weight, age and sex. In a competition, two kickboxers of the same weight and the same age class compete for points awarded in specified target areas in the form of punches and kicks, or for a technical knockout (KO). Success in kickboxing requires a combination of technical and tactical skills as well as mental, physical, and emotional preparation (Buse & Santana, 2008).

In order to optimize performance, athletes and their coaches use different types of ergogenic preconditioning strategies. Depending on the type of sport, coaches and athletes use different strategies or aids. One of them is music, which has proven to be effective and used in several sports (Ouergui et al., 2023a; Jebabli et al., 2023b; Blasco-Lafarga et al., 2022). In fact, the benefits of listening to music includes delayed perception of neuronal fatigue (Diehl et al., 2023), improved muscle efficiency (Centala et al., 2020), increased neuronal activity (Bigliassi et al., 2017), and improvements in mood (Jebabli et al., 2023a), attention (Patania et al., 2020), and self-efficacy (Pettit & Karageorghis, 2020). The ergogenic potential of music, when used during exercise or as a pre-exercise intervention, is often limited by regulatory frameworks, as most competitive sports do not allow the use of auditory stimuli during exercise. Listening to music during warm-up therefore presents itself as a plausible strategy to improve performance outcomes in competitive contexts. Also, to maximize music benefits, previous studies reported that self-selection of upbeat music (preferred music) with personally emotive qualities is worthy of consideration (Terry et al., 2020).

While several findings have been documented in many sports, the use of music’s ergogenic properties in kickboxing remains a relatively unexplored territory. However, this gap has been gradually filled by recent studies that investigated the impact of music in some other combat sports such as kickboxing (Boujabli et al., 2024) and taekwondo (Ouergui et al., 2023a; Ouergui et al., 2023b; Delleli et al., 2024; Messaoudi et al., 2024). For example, Boujabli et al. (2024) observed that listening to preferred music with or without video feedback during warm-up improve positive feeling and repeated roundhouse kick physical performance. However, the application of music to specific kickboxing tasks, including combined punches and kicks, has not been studied. Also, Ouergui et al. (2023a) showed that the use of pre-selected music during warm-up in taekwondo athletes would improve mood satisfaction and physical performance. Ouergui et al. (2023a) observed that listening to music with a tempo of 140 beats per minute, a sound volume of 80 decibels was the most favorable condition to obtain better physical performance as well as better punching speed. In addition, previous research suggests that it is beneficial to listen to music during warm-up phases in taekwondo (Delleli et al., 2024; Messaoudi et al., 2024). Nevertheless, it remains ambiguous as to whether such effects can be carried over to kickboxing athletes.

In addition, previous studies show that any ergogenic effect of music in physical performance depends on auditory characteristics such as the type of music, tempo, volume, duration, or the timing of music exposure (Karageorghis & Priest, 2012; Jebabli et al., 2022; Jebabli et al., 2023a). These studies reported that the optimal effect of music on physical performance relies on specific characteristics, including high musical volume (80 dB) and fast tempo (≥120 BPM) (Jebabli et al., 2022; Jebabli et al., 2023a; Jebabli et al., 2023b; Karageorghis et al., 2018; Ouergui et al., 2023a; Ouergui et al., 2023b). Despite these results, we do not know exactly the relational effect between musical tempo and frequency on athletic performance and psychophysiological responses. In other word, the synergistic integration between frequency and rhythm of music remains unknown. However, music tuned to different frequencies such as 440 Hz and 432 Hz has attracted attention for its distinct positive effects on public health (Calamassi & Pomponi, 2019; Halbert et al., 2018; Suarez et al., 2024).

Theoretically, the basic source of musical sound is given by the frequency in hertz (Hz) (Gray, 1999). These frequencies theoretically define the pitch and timbre of the sound produced (Rutherford-Johnson, 2017). The frequency around 440 Hz or the pitch of A-440 Hz has been accepted as a standard reference for tuning many musical instruments. In fact, in 1975, ISO had already issued 440 Hz as the standard tuning frequency for most musical compositions. This frequency corresponds to pitch class A4. Thus, such standardization has become an international reference point in musical tuning, influencing a wide range of performances and musical records.

Music at 440 Hz with fast tempos and standards frequencies is crisp and clear, increasing concentration and intensity in workouts, anaerobic performance by increasing heart rate and perceived exertion, especially during high-intensity training sessions (Ouergui et al., 2023a; Ouergui et al., 2023b; Jebabli et al., 2023a). On the other hand, music at 432 Hz is generally considered more harmonious and relaxing; recently, it has also begun to be introduced during recovery and mental preparation phases, with claims of reducing anxiety, improving relaxation and increasing the mind-body connection (Fauble, 2016). However, no studies have compared the effects of 440 Hz vs. 432 Hz music in physical performance, an area in which research is particularly worthy of interest.

Therefore, the aim of the present study was to investigate the effects of listening to preferred music at different frequencies (i.e., 440 Hz vs. 432 Hz) during warm-up on specific physical performance in kickboxing. Based on the above objectives, it was hypothesized that: (a) listening to preferred music at a fast pace with frequencies around 440 Hz and 432 Hz improves specific physical performance in kickboxing for both sexes; (b) listening to preferred music at 432 Hz is more effective in reducing RPE compared to the preferred music condition at 440 Hz; (c) the effect of preferred music on physical performance has the same degree of improvement for both sexes.

Materials & Methods

Study design

This study is a randomized, double-blind, crossover trial examining the impact of listening to preferred music at frequencies of 440 Hz or 432 Hz during warm-up on physical performance and psychophysiological responses in kickboxing. Kickboxers were exposed to three conditions: (1) listening to preferred music around 440 Hz (PM-440 Hz) during warm-up, (2) listening to preferred music around 432 Hz (PM432 Hz) during warm-up, and (3) neutral self-talk with no-music during warm-up (control condition). Prior to the experimental procedures, the kickboxers were thoroughly familiarized with the testing protocols. Each athlete completed the Intermittent Kickboxing Anaerobic Speed Test (IKAST) (Gençoğlu et al., 2023) under each condition in separate sessions, with a 48-hour recovery period between sessions. Previous studies have shown that a 48-hour recovery period was sufficient to restore neuromuscular function, normalize muscle strength, and physiological adaptations that prepare the body for future sessions (Milioni et al., 2021; Miranda et al., 2018).

Before each testing session, kickboxers underwent a standardized 10-minute warm-up protocol according to Van Den Tillaar, Lerberg & Heimburg (2019), which included 5 min jogging (60–65% of maximal heart rate), 4 min of running (three runs of 60 m at 75, 85 and 95% of maximal self-estimated intensity; recovery: 1 min between each run), and one minute of lateral movements and dynamic stretching. After this, two minutes of passive recovery was followed before the specific warm-up was continued under one of the three conditions. All testing sessions were conducted at the same time of day (3PM ± 1 h) to control for diurnal variations in performance and in the same gym with a moderate temperature (23–25 °C).

Kickboxers were instructed to avoid vigorous exercise for 48 h prior to each testing session. They were also advised to maintain their usual hydration, dietary habits, and sleep patterns, and to refrain from consuming any ergogenic supplements (e.g., caffeine, vitamins) in the 24 h leading up to each session.

The study’s design and progression are depicted in Fig. 1, which provides a Consolidated Standards of Reporting Trials (CONSORT) flow diagram, and Fig. 2 details the methodological rigor.

Figure 1 Diagram depicting the consolidated standards for reporting trials in the study.

Note: PM-440 Hz, listening to preferred music around 440 Hz; PM432 Hz, listening to preferred music around 432 Hz.

Figure 2 Study design.

Kickboxers

A priori power analysis was conducted using G*Power software (Version 3.1.9.4; University of Kiel, Kiel, Germany) with the F test family (ANOVA: repeated measures, within factors). The sample size calculation, using G*Power 3.1.9.7 software (Franz Faul, University of Kiel, Kiel, Germany), was based on a statistical power of 0.80, a significance level of 0.05, and an effect size of 0.25 taken from a related study from a similar kicking test (d = 0.48) (Ouergui et al., 2023a; Ouergui et al., 2023b). The analysis indicated that 28 kickboxers were needed to achieve 80% power.

Twenty-eight kickboxers (15 men and 13 women) volunteered to participate in the study (Table 1). All kickboxers were recruited from the same local training club and met the following inclusion criteria: (1) at least 4 years of kickboxing experience; (2) no muscular or joint injuries; and (3) for women, no menstrual-related dysfunctions (such as amenorrhea) and no use of hormonal contraception in the 2 months prior to the study.

Table 1 Characteristics of the participants.

Sex	N	Age
(years)	Body mass (kg)	Height
(m)	BMI
(kg m−2)	Kickboxing
experience
(years)	
Men	15	19.53 ± 2.23	67.63 ± 7.91	1.78 ± 0.08	21.36 ± 1.05	4.47 ± 0.9	
Women	13	18.08 ± 1.12	56.71 ± 11.53	1.63 ± 0.09	21.15 ± 2.99	3.96 ± 0.9	
Overall	28	18.86 ± 1.92	62.56 ± 11.06	1.71 ± .11	21.27 ± 2.13	4.23 ± 0.92	
Notes.

Values are mean ± standard deviation (n = 28).

After a thorough explanation of the study’s objectives and potential risks, athletes provided written informed consent. The study adhered to the most recent Declaration of Helsinki guidelines for human research and received approval from the local ethics committee at the High Institute of Sport and Physical Education of Kef on March 9, 2022 (approval number UR22JS01/ISSEP-015-22) prior to the commencement of data collection.

Musical characteristics

During familiarization sessions, kickboxers were asked to preselect their preferred music to be played during the test in the experimental protocol sessions. Using the Audacity application, the tempo of each chosen song was adjusted to a fast tempo of 140 beats per minute (bpm) and set to a volume of 80 dB. Each song was played for 10 min per session, during warm-up, through the same wireless headphones (AirPods Pro, Apple, US) for all kickboxers. The music was recorded at two different frequencies, 440 Hz and 432 Hz in WAV format using the Audio Processing Object (APO) software. To ensure the double-blind procedure, the recordings at different frequencies (440 Hz; 432 Hz) were made by an independent researcher who was not directly involved in the present study. During the no-music condition, headphones were worn, but no music was played.

Measurements

The specific anaerobic speed test in kickboxing (IKAST) involved five repetitions of a combination of four techniques: (1) right-left punch, (2) right roundhouse kick, (3) right-right punch, and (4) left roundhouse kick. This sequence was performed over five sets with a 10-second rest between each set (Gençoğlu et al., 2023). The total execution time of the test was used as a measure of physical performance.

Performance indices assessed included: total time of sequences performed (total time), fastest sequence time performed presented as the best repetition (peak time) and fatigue index. The fatigue index, indicating the relative decrease in power, was calculated using the following formula: Fatigue index%=1−peak time×5/total time×100.

Additionally, the highest speed of the best technical impact was determined. All physical indices (duration and maximal speed) during the test were analyzed from video recordings using Kinovea software (version 0.9.5). Videos were recorded at a resolution of 1,080 p (1,920 × 1,080, 16:9) and 48 frames per second (FPS) using the GoPro4 session camera.

During the test, a heart rate monitor (Polar Team 2; Polar Electro Oy, Kempele, Finland) recorded both peak heart rate (HRpeak) and mean heart rate (HRmean). Kickboxers’ overall physical exertion was assessed immediately after the test using the Rating of Perceived Exertion (RPE) scale (6–20 Borg scale; Borg, 1982). Additionally, the feeling scale was used to assess affective responses by using an 11-point bipolar numeric rating scale (ranges from −5 (very unpleasant) to +5 (very pleasant)) measuring current mood after testing (Hardy & Rejeski, 1989). Three minutes after the test, blood samples were taken from the fingertip (five µl of blood) to measure blood lactate concentrations using a portable lactate monitor (lactate pro2; Arkray, Kyoto, Japan).

The intra-class correlation coefficients (ICC) for test–retest reliability were 0.91 for total time, 0.89 for best time and 0.87 for fatigue index.

Statistical analysis

Normality of data distribution was assessed and confirmed using the Shapiro–Wilk test. Test–retest reliability for all variables was evaluated using Cronbach’s intraclass correlation coefficient (ICC) and the coefficient of variation (CV).

To investigate the effect of condition and sex, a 2 (sex) × 3 (test condition: repeated measures) ANOVA was conducted to analyze differences among conditions for average and peak hear rates, blood lactate, maximal speed impact, total and best set time, fatigue index, RPE and feeling scale. To assess difference between the sets of the IKAST and conditions a 2 (sex) × 3 (test condition) × 5 (set 1–5) ANOVA of repeated measures was conducted. When significant differences were detected, post-hoc comparisons were performed with Holm-Bonferoni correction. Effect size was evaluated with Eta partial squared where 0.01 < η2 < 0.06 constitutes a small effect, 0.06 < η2 < 0.14 a medium one and η2 > 0.14 a large effect (Cohen, 1988). Where the sphericity assumption was violated, the Greenhouse–Geisser adjustments of the p-values were reported. The level of significance was set at p < 0.05. All data analyses were performed using JASP v. 0.17.3 (University of Amsterdam, Amsterdam, Netherlands). Data were presented as means and standard deviations (SD).

Results

A significant interaction effect on maximal impact velocity, feeling scale and mean heart rate were found. Also, a significant main effect of sex was observed for total time, best time and highest impact speed (F ≥ 21.4, p < 0.001, η2 ≥ 0.45), while significant main effect of condition was found upon all parameters (F ≥ 21.0, p < 0.001, η2 ≥ 0.15). However, no significant main effect of condition was observed for fatigue index (F = 0.57, p = 0.57, η2 = 0.01) and blood lactate concentration (F = 2.0, p = 0.145, η2 < 0.01).

Post-hoc comparisons revealed that men were faster in the best set and total time and had a higher impact speed than women in all conditions (Table 2). After the warm-up with PM440 Hz, both men and women performed the IKAST significantly faster in the best time and total time indices with highest feeling scale and lowest RPE followed by the warm-up with PM432 Hz for only the feeling scale compared to control condition. Furthermore, a significant interaction effect on the feeling scale was observed as women had a significantly higher feeling scale after PM440 Hz condition compared to the PM432 Hz condition, which men did not have, while the opposite was found in the maximal impact speed where men had a significantly higher speed after PM440 Hz compared to the PM432 Hz condition and women did not. In addition, the control condition had a significantly higher mean heart rate and peak heart rate of the warm-up and a lower maximal impact speed compared to and PM440 Hz. Also, the mean heart rate was significantly the highest during the control condition, while no significant difference between the other two conditions was found, which was caused by the interaction effect between sex in the PM440 Hz and PM432 Hz conditions: men had a significant lower mean heart rate after PM440 Hz condition compared to PM432 Hz condition, while the women had the lowest mean heart rate after the warm-up with PM432 Hz condition compared with PM440 Hz condition (Table 2).

Table 2 Effects of listening to preferred music at frequencies around 440 Hz or 432 Hz and no music during warm-up on physical performance and psychophysiological responses on men and women in kickboxing.

Parameter	Sex	No music	440 Hz	432 Hz	Condition	Sex	Condition ×sex	
Total time (s)	Men	50 ± 2.65†	46.78 ± 2.66†	47.87 ± 2.60†	<0.001 (0.08)	<0.001* (0.43)	0.82 (<0.01)	
Women	53.66 ± 2.27	51.26 ± 2.26	52.35 ± 2.20	
Both	51.63 ± 3.33*	48.86 ± 3.33*	49.95 ± 3.29*	
Best time (s)	Men	9.72 ± 0.51†	9.24 ± 0.52†	9.47 ± 0.50†	<0.001* (0.09)	<0.001* (0.42)	0.70 (<0.01)	
Women	10.58 ± 0.46	10.11 ± 0.46	10.32 ± 0.44	
Both	10.11 ± 0.65*	9.64 ± 0.65*	9.87 ± 0.67*	
Fatigue index (%)	Men	−1.25 ± 0.37	−1.26 ± 0.35	−1.03 ± 0.41	0.56 (0.01)	0.061 (−0.51)	0.38 (0.02)	
Women	−1.40 ± 0.34	−1.39 ± 0.33	−1.43 ± 0.77	
Both	−1.31 ± 0.36	−1.32 ± 0.34	−1.28 ± 0.46	
Highest impact speed (km/h)	Men	41.59 ± 2.73†	42.24 ± 2.69†	42.18 ± 2.69†	<0.001* (<0.01)	<0.001* (0.45)	<0.001* (<0.01)	
Women	37.01 ± 2.63	37.48 ± 2.64	37.48 ± 2.64	
Both	39.47 ± 3.51*	40.03 ± 3.57	40.00 ± 3.54	
Mean heart rate (beats/min)	Men	168 ± 5.2‡	165 ± 5.17‡	166.0 ± 5.2‡	<0.001* (0.06)	0.81 (<0.01)	<0.001* (0.01)	
Women	168.5 ± 4.4‡	166.4 ± 4.5‡	165.46 ± 4.4‡	
Both	168.2 ± 4.8*	165.6 ± 4.8	165.8 ± 4.8	
Peak heart rate (beats/min)	Men	183.1 ± 3.4	182.4 ± 2.5	180.1 ± 3.4	<0.001* (0.15)	0.93 (<0.01)	0.23 (<0.01)	
Women	183.8 ± 3.1	181.5 ± 2.9	180.6 ± 3.2	
Both	183.4 ± 3.2*	182.0 ± 2.6*	180.4 ± 3.2*	
Lactate (mmol/L)	Men	5.74 ± 0.73	5.68 ± 0.63	5.70 ± 0.65	0.15 (<0.01)	0.67 (<0.01)	0.99 (<0.01)	
Women	5.62 ± 0.91	5.55 ± 0.89	5.58 ± 0.92	
Both	5.69 ± 0.80	5.62 ± 0.75	5.64 ± 0.77	
Feeling scale (−5 to +5)	Men	2.4 ± 0.63‡	3.73 ± 0.80	3.80 ± 0.94	<0.001* (0.45)	0.47 (<0.01)	<0.001* (0.05)	
Women	2.15 ± 0.69‡	4.15 ± 0.69‡	3.08 ± 0.76‡	
Both	2.29 ± 0.66	3.93 ± 0.77	3.46 ± 0.92	
RPE (1–10)	Men	8.40 ± 0.83‡	6.87 ± 0.92‡	7.47 ± 1.19‡	<0.001* (0.33)	0.29 (0.02)	0.27 (<0.01)	
Women	8.23 ± 0.83‡	6.62 ± 0.87	6.85 ± 1.07	
Both	8.32 ± 0.82	6.75 ± 0.89	7.18 ± 1.16	
Notes.

Data are shown as the mean (standard deviation); P value (partial squared).

* Indicates a significant difference with all other conditions for both men and women (p < 0.05).

‡ Indicates a significant difference with other conditions for this sex (p < 0.05).

† Indicates a significant difference between men and women (p < 0.05).

When evaluating the development over the five sets of the IKAST between the conditions, a significant condition × set (F = 21.2, p < 0.001, η2 < 0.01), set × sex (F = 7.2, p < 0.001, η2 < 0.01) and set × sex × condition (F = 2.9, p = 0.005, η2 < 0.01) interactions were found. Also, a significant main effect of set (F = 26.7, p < 0.001, η2 = 0.01), main effect of condition, (F = 59.82, p < 0.001, η2 = 0.08) and main effect of sex (F = 23.1, p < 0.001, η2 = 0.42) were found.

Post-hoc comparisons revealed that the set times were significantly longer when using the control condition in each set followed by PM432 Hz and the shortest sets at the PM440 Hz condition. In the control and PM440 Hz conditions, the time increased from set to set when evaluated for all subjects together, but only from set 2 to 3 in PM432 Hz, followed by a decease and increase in set 4 and 5 for this condition (Fig. 3A). Furthermore, men were faster than women in all sets in all conditions in which both men and women increased times from set to set in control condition. While women also did this in the PM440 Hz condition, men did not significantly increase the time between set 4 and 5. Furthermore, in the PM432 Hz condition, both groups increased from set 2 to 3 and between set 4 and 5, while the women showed a significant decreased time from set 3 to 4 and men kept the same time between these two sets (Fig. 3B).

Figure 3 Mean (±SEM) effects of listening to preferred music at frequencies around 440 Hz or 432 Hz and no music during warm-up per set for (A) all subjects and (B) separated for men and women.

Note: An asterisk (*) indicates a significant difference between the conditions for all sets; † indicates a significant difference between men and women for all conditions and sets (p < 0.05); → indicates a significant difference between this set with the next one (p < 0.05).

Discussion

The primary aim of this study was to assess the effects of listening to preferred music at different frequencies (440 Hz vs. 432 Hz) during warm-up on specific physical performance and psychophysiological responses in amateur kickboxers. The main findings of this study were that after the warm-up with PM440 Hz both men and women performed the IKAST significantly faster (shorter total and best times) with positive mood and lowest RPE followed by the warm-up with PM432 Hz and thereafter significantly again by the control condition. Furthermore, the control condition had significantly higher mean and peak heart rate of the warm-up and a lower maximal impact speed. For the sex interaction, men had a lower heart rate during PM440 Hz and women had the lowest heart rate during PM432 Hz. Additionally, women had a higher positive feeling mood during PM440 Hz, while men did not experience a significant change between the two music conditions.

The results of the present study indicated that warm-up with PM440 Hz resulted in the fastest IKAST performances compared with PM432 Hz and control conditions in both men and women kickboxers. Given that most musical compositions globally use a standard frequency of 440 Hz as a tuning reference and by the fact that kickboxers are generally exposed from birth to music whose vibrations and harmonics are determined exclusively at 440 Hz. This makes this frequency more “familiar” to them as shown by the best improvement upon IKAST performance (Table 2 and Fig. 1). This was in accordance with improvement in repeated sprint performances after listening to the preferred music of 440 Hz during warm-up (Chtourou et al., 2012; Eliakim et al., 2007; Ouergui et al., 2023a). For example, Ouergui et al. (2023a) found that listening to pre-selected music during warm-up at a volume of 80 dB and a fast tempo of 140 beats/minute improves kicking performance in taekwondo male and female athletes, by increasing the number of circular kicks performed during 10-second repeated kicking tests. Those results are likely attributable to several factors including enhanced muscle efficiency (Centala et al., 2020), increased neuronal activity (Bigliassi et al., 2017), greater self-efficacy (Pettit & Karageorghis, 2020) and improved attention (Patania et al., 2020).

This focus may have shifted their attention from the music to internal cues, including proprioception and kinesthesia in order to improve physical performance (Manca et al., 2020). Additionally, listening to music may offer improved distraction from fatigue-related symptoms by modulating beta frequency activity in the brain (Bigliassi et al., 2019) which influence mental state during intense physical activity. However, no such effect was observed in the present study, as indicated by the absence of a significant change in the fatigue index.

Previous research has demonstrated, for therapeutic purposes, that 432 Hz music reduces stress, anxiety and enhancing sleep quality in patients with various chronic diseases (Calamassi & Pomponi, 2019; Di Nasso et al., 2016; Dubey et al., 2019). From these findings, we can see that the relaxing effect of 432 Hz music does not have a stimulating effect on attention, which could limit its impact on improving anaerobic performance.

From a physiological perspective, the present study showed both lower peak and mean heart rates during IKAST in PM432 Hz and PM440 Hz conditions compared to no music condition in men and women kickboxers. The physiological mechanisms associated with listening to music at different frequencies are unclear. However, a possible reason for our results is that PM432 Hz and PM440 Hz can affect the autonomic nervous system by focusing attention on the external environment rather than on bodily sensations or perception of unwanted sensations (Maddigan et al., 2019). This can quickly trigger a parasympathetic response, thus accelerating heart rate recovery during recovery (Maddigan et al., 2019). The lower heart rates support our findings that the parasympathetic nervous system and the peripheral nervous system can improve physical performance during high-intensity intermittent exercise through physiological regulation, which in accordance with the review by Terry et al. (2020). They found showed that music clearly improves performance when it results in reduced heart rate values even when workload is maintained or increased. Furthermore, few studies have suggested that low-frequency music may have a potential effect on reducing heart rate and promoting relaxation in sedentary individual, even in the absence of physical exercise (Calamassi & Pomponi, 2019; Halbert et al., 2018).

Regarding the sex interaction, men had lower heart rates during the PM440 Hz condition while women had lower heart rates during the PM432 Hz condition. These heart rate responses associated with listening to music at different frequencies for each sex were very surprising. However, the heart rate variation in men at rest is less pronounced than in women, indicating a potential difference in autonomic regulation during the music condition (Goshvarpour et al., 2013).

Despite the increases in physical performance during PM440 Hz, blood lactate levels measured three minutes after IKAST were not significant different between conditions indicating that the anaerobic processes were similar between conditions. While other studies reported that music can enhance muscle blood flow and thus affect lactate clearance during high-intensity exercise (Ghaderi et al., 2015; Jebabli et al., 2023a), the discrepancy in results could be attributed to differences in the types of exercise, music characteristics and the type of participants.

From a psychological perspective, the present study showed that listening to music during warm-up, regardless of the frequency, improves positive mood with lower ratings of perceived exertion (RPE) compared to no-music condition. These findings are consistent with previous researches demonstrating the reducing effect of music on RPE (Ballmann et al., 2019; Ballmann, 2021). This phenomenon may be due to enhanced dissociation from discomfort and exertion during exercise following exposure to preferred music during the warm-up phase. Music is well-established as an effective external distractor that reduces sensations of fatigue (Ballmann et al., 2019; Potteiger, Schroeder & Goff, 2000; Bigliassi et al., 2019). For instance, Ballmann et al. (2019) found that RPE was lower in kickboxers who performed the Wingate test while listening to their preferred music.

Besides the positive effect of listening to music in both sexes on mood state, our results showed that the frequency difference only influenced the women as they had a significantly more positive feeling mood after listening to PM440 Hz music compared with PM432 Hz during the warm-up. While men had a reduced RPE after PM440 Hz condition compared to PM432 Hz condition, women had similar reductions after both music warm-ups which was in accordance with Rhoads et al. (2021) who reported that women had lower RPE values during music conditions compared to the control condition. These different responses between sexes can perhaps be explained by the evidence that women tend to show greater emotional responses to musical stimuli than men (Nater et al., 2006). It is not yet clear how these differences arise, but there are indications that there are sex differences in how the brain responds to music (Koelsch et al., 2003). These small but important differences in brain activation during exercise with music may allow women to better dissociate physical and psychological fatigue. Consistent with this hypothesis, Carlson et al. (2015) found that women’s prefrontal cortex activity while listening to music better maintained attention to negative thoughts than men.

Other sex differences found in the present study were the faster execution time and higher maximal impact speed of men compared to women. This was expected, as men have been shown to be stronger and faster than women (Nikolaidis et al., 2016). However, the development in execution times over the sets were similar for the men and women kickboxers (Fig. 1) indicating that the warm-ups have similar effect on the physical performance in both sexes. Although the maximal impact speed did not significantly increase in the women while it did in men, causing a significant interaction effect, this had only a very small effect size (0.01) and was therefore not so important. The same was found for mean heart rates in which a significant interaction effect was found as both reduced mean heart rates after a warm-up with listening to music, but men had the lowest heart rate after PM432 Hz condition, while women had the lowest heart rate after PM440 Hz warm-up. The difference was only on average one beat/min and had a very small effect size and thereby not so important.

Finally, we acknowledge some limitations of the present study. Indeed, the absence of indices such as neural activation that provide more details on how each sex responds to exercise after listening to preferred music at different frequencies. Similarly, the IKAST applied in this study was an anaerobic test specific to kickboxing that does not reflect the global and cumulative psychophysiological stress developed by kickboxers during competition. In this context, further studies are needed to determine the effect of preferred music at different frequencies on physical and psychophysiological responses during a simulated kickboxing combat.

Conclusions

Listening to preferred music with at high tempo (140 beats/min), high (80 dB) loudness with frequencies of 440 Hz and 432 Hz are both effective for kicking performance in men and women kickboxers. However, the standard frequency around of 440 Hz applied during warm-up was more effective to improve specific performance and positive mood with potential dissociation from discomfort during test, with some sex difference in positive mood as women were affected by the difference in frequency and men were not affected. The practical results of these findings highlight the benefits of listening to music in warm-up has been shown to improve physical performance and positive mood while decreasing subjective fatigue; therefore, coaches have a practical approach to help athletes better prepare for competition and make the training experience more enjoyable and pleasant.

Supplemental Information

Supplemental Information 1 Data of participants

The authors would like to thank all subjects for their participation in the study.

Additional Information and Declarations

Competing Interests

Author Contributions

Human Ethics

Data Availability

The authors declare there are no competing interests.

Nidhal Jebabli conceived and designed the experiments, performed the experiments, prepared figures and/or tables, authored or reviewed drafts of the article, and approved the final draft.

Manar Boujabli conceived and designed the experiments, authored or reviewed drafts of the article, and approved the final draft.

Mariem Khlifi conceived and designed the experiments, performed the experiments, authored or reviewed drafts of the article, and approved the final draft.

Nejmeddine Ouerghi conceived and designed the experiments, performed the experiments, authored or reviewed drafts of the article, and approved the final draft.

Anissa Bouassida conceived and designed the experiments, authored or reviewed drafts of the article, supervisor, and approved the final draft.

Abderraouf Ben Abderrahman conceived and designed the experiments, authored or reviewed drafts of the article, supervisor, and approved the final draft.

Roland van den Tillaar analyzed the data, prepared figures and/or tables, authored or reviewed drafts of the article, supervisor, and approved the final draft.

The following information was supplied relating to ethical approvals (i.e., approving body and any reference numbers):

Local ethics committee at the High Institute of Sport and Physical Education of Kef Tunisia (Ethical Application Ref: UR22JS01/ISSEP-015-22).

The following information was supplied regarding data availability:

The raw measurements are available in the Supplementary File.

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
