# Peer review of "Effects of 440-Hz vs. 432-Hz preferred music frequencies, during warm-up, on intermittent anaerobic speed test performance in men and women kickboxers: a double-blind crossover study"

_PeerJ, doi:10.7717/peerj.19084_

## Round 0.1 · original submission · Major Revisions

Dear authors,

The reviewers have offered several suggestions concerning the clarity of the text and the presentation of the results. A thorough review of these issues is necessary prior to a subsequent evaluation. I would like to highlight that one of the reviewers submitted an annotated manuscript for your consideration. Regards.

Reviewer 1 ·

Basic reporting

English is quite clear, references are quite sufficient to explain the background.
The article structure is okay. However, I suggest to report the subheadings in the abstract.

Experimental design

The experimental approach is well-defined. However, I have highlighted some points of improvement in the points below.

Validity of the findings

Data may give potential implications in terms of strategies to enhance athletes' performance.

Additional comments

Abstract:
Please add the subheadings (Background/Materials etc.)

Introduction:
Some relevant references on this topic are missing (10.3390/sports12050131; 10.1016/j.jsams.2024.01.002). Moreover, as the main administration is regarding the preferred pre-task music, an addition of this conceptual framework behind the aim should be briefly explained. As several domains of the effects of music are investigated (see Terry et al. 2020), they should be better clarified in the introduction section.

Material and methods:
Line 125: Are you sure that "48-hour recovery period between sessions" was enough to avoid fatigue between sessions? Please justify this aspect.
Figure 2 should be more explicative and not resume the text of the manuscript.
Line 146: How the effect size was estimated?
Line 149: Did you ask about participants' previous experience in listening to music during exercise?
Line 158: How were asked the participants to choose their preferred music?
Participants also wore the hearphones in the no music condition?

Discussion:
The importance of the feeling scale should be mentioned in the discussions. According to Karageorghis, music could have an effect in different domains. I suggest to discuss the results according to each domain (e.g., psychological, performance etc).

English: I suggest rechecking some grammars spellings throughout the manuscript.
Be consistent with the use of "sex" or "gender" throughout the manuscript.

·

Basic reporting

The study aims to evaluate the effects of preferred music on physical performance in a group of kickboxers. The hypotheses seem descriptive rather than inferential. The authors have included several dependent variables (See table 2) of interest but lost themselves into details that might hinder the main message. The outcomes of the study might support previous studies showing the impact of exercise of hedonic responses. The authors might consider tailoring their discussion towards the affective / hedonic response to exercise and its related mechanisms.

Experimental design

The double-blind cross-over study design confers to the study its strength. We must commend the authors for their effort to have implemented this rigorous experimental design. That said, the control group should have been assigned a task (neural) to avoid a potential Hawthrone effect. The methodology well suits the research question and the statistical power is clearly sufficient.

Validity of the findings

The results of the study represent the strongest part of the manuscript. However, the reviewer considers that the authors did not extract the full potential of their outcomes. The reviewer thinks that the authors should discuss their major outcomes within the theoretical framework of affective / hedonic aspect of exercise. The results of the study are in line with psychophysiology, which is currently attempting to establish the interaction between peripheral sensations and central integration before, during and after exercise.

Additional comments

I have uploaded the manuscript with some comments.

---

## Round 0.2 · accepted · Accept

Dear authors, thank you for addressing the concerns raised. Reviewers and I consider the manuscript ready for publication. Congratulations on your work.

Reviewer 1 ·

Basic reporting

Authors addressed my comments and I think that now the article is suitable for publication.

Experimental design

no comment

Validity of the findings

no comment

Additional comments

no comment

·

Basic reporting

All criteria have been met in the resubmission

Experimental design

The authors have clarified some important aspects of their methodology.

Validity of the findings

Now that authors have rewritten the major parts of the discussion, the paper should have an impact in the field.

Additional comments

Very good piece of work.